# Physiological Response to Nitrogen Deficit in Potato Under Greenhouse Conditions

**DOI:** 10.3390/plants14213237

**Published:** 2025-10-22

**Authors:** Leire Barandalla, Alba Alvarez-Morezuelas, Carmen Iribar, Enrique Ritter, Patrick Riga, Maite Lacuesta, Jose Ignacio Ruiz de Galarreta

**Affiliations:** 1NEIKER-Basque Institute for Agricultural Research and Development, Basque Research and Technology Alliance (BRTA), Campus Agroalimentario de Arkaute, 01080 Arkaute, Spain; lbarandalla@neiker.eus (L.B.); a.alvarez.morezuelas@gmail.com (A.A.-M.); carmen.iribar@neiker.eus (C.I.); eritter@neiker.eus (E.R.); 2NEIKER-Basque Institute for Agricultural Research and Development, Basque Research and Technology Alliance (BRTA), Parque Tecnologico de Bizkaia, C/Berreaga, 48160 Derio, Spain; priga@neiker.eus; 3Grupo FisioKlima AgroSosT, Facultad de Farmacia, UPV/EHU, Pº de la Universidad, 7-01006 Vitoria-Gasteiz, Spain; maite.lacuesta@ehu.eus

**Keywords:** nitrogen deficit, *Solanum tuberosum* L., yield, chlorophyll

## Abstract

Potatoes have low nitrogen (N) use efficiency due to their shallow root systems, which results in nitrate loss and reduced yield. The objective of this study was to determine physiological parameters, yield, and quality components in three potato varieties subjected to N deficiency in a greenhouse, as there are few previous studies on the relationship between N deficiency and physiological parameters. The control plants were fertilized twice with 27% calcium ammonium nitrate, while the stressed plants were not fertilized. Chlorophyll content (SPAD), biomass, N, and leaf area showed highly significant differences (*p* value < 0.05) between the control and stressed plants, the latter showing higher photosynthetic activity. Agria cv. obtained the highest SPAD value (T0: 47.93, T1: 44.45; T2: 40.18) under stress. In tubers, the concentrations of N, amino acids, vitamin C, and phenols were higher in the control plants, and Agria exhibited the greatest reduction under stress conditions. Production was reduced the most in Kennebec, with 29.22%, compared to Agria with 15.73% and Monalisa with 26.58%. The Agria variety under N deficiency showed the lowest values of nutritional compounds such as vitamin C. Physiological parameters such as photosynthetic activity showed significant correlations with tuber quality parameters such as vitamin C, amino acids, and macro- and microelements. These parameters may be useful for stress identification, as well as for the selection of more N-deficiency-tolerant parents in potato breeding programs. Upcoming studies and investigations will seek to validate the parameters assessed in field trials.

## 1. Introduction

Nitrogen (N) is a critical nutrient for crop growth and development, particularly for potatoes (*Solanum tuberosum* L.), the world’s fourth most important food crop after rice, wheat, and maize [1]. Adequate N fertilization enhances leaf expansion, promotes canopy formation, and improves photosynthetic efficiency and the distribution of dry matter to the tubers, contributing to their bulking and overall yield [2,3,4]. For industrial potatoes, starch content and tuber yield are particularly important and are strongly influenced by water and N availability [5]. However, excessive N fertilization can lead to overgrowth of stolons and leaves, which can hinder tuber development, maturity, and quality [6].

However, excessive N application produces serious problems and environmental damage. Over-fertilization can lead to excessive vegetative growth, delayed tuber development, and reduced product quality. Thus, adequate N fertilization is fundamental to establishing sustainable agricultural systems. Nowadays, more than half of the 120 million tons of N fertilizer applied is lost to air and water, contributing to air pollution, water pollution, soil acidification, climate change, ozone depletion in the stratosphere, and biodiversity loss [7,8,9].

Potatoes are known to have a relatively low N uptake efficiency, ranging from 50% to 60% [10], because their shallow root systems are less effective at absorbing N compared to other crops, like wheat, maize, or sugar beet [11]. Around 85% of the total root length is concentrated in the upper 0.3 m of the soil [12], which is detrimental to both water use efficiency (WUE) and nitrogen use efficiency (NUE). The average N recovery by plants is 37%, although it varies from species to species and depends on climate and crop management [13].

Despite recommended fertilization rates ranging from 70 to 330 kg N/ha, there is a lack of comprehensive studies that integrate physiological, agronomic, and biochemical responses to N fertilization in potatoes. Improving nitrogen use efficiency (NUE) is especially important in this context. NUE is genotype-dependent and involves two distinct processes: nitrogen uptake efficiency (NUpE), the plant’s ability to extract N from the soil, and nitrogen utilization efficiency (NUtE), its capacity to convert absorbed N into yield [14,15]. Selecting genotypes with high NUE is critical for sustainable production, particularly under nitrogen-limited conditions. Developing cultivars with enhanced agronomic performance requires a thorough assessment of genetic resources, focusing on yield and its components under varying N availability [1].

Recent studies have revealed that nitrogen deficit triggers complex physiological and molecular responses in plants, including changes in root structure, chlorophyll reduction, and activation of stress-related genes [16]. In potatoes, nitrogen scarcity is genetically linked to SNP markers on chromosomes 2 and 3, influencing traits like leaf area, chlorophyll content, and tuber number [17]. Proteomic analysis under nitrogen stress reveals altered amino acid metabolism with genotype-specific patterns, and nitrogen scarcity has been shown to enhance stolon growth and tuber formation by upregulating starch and sucrose pathways [18,19].

Despite these insights, there is a lack of integrated studies that examine physiological, agronomic, and biochemical responses of different potato varieties under nitrogen deficit—especially in controlled greenhouse conditions. This study hypothesizes that potato varieties with distinct vegetative cycles exhibit genotype-specific responses to nitrogen stress.

The aim of this study was to evaluate the physiological, agronomic, and biochemical responses of three potato varieties with different vegetative cycles under nitrogen-deficient conditions to identify selection parameters useful for breeding programs.

## 2. Results

The experiment involved planting 20 tubers per variety under a completely randomized design. Each variety was split into two treatments: ten pots received nitrogen-deficit stress (D), while ten were irrigated normally (C).

### 2.1. Physiological Responses to Nitrogen Deficit

Highly significant differences (*p* ≤ 0.0001 and *p* ≤ 0.001) between treatments (control vs. N-deficient plants) in parameters such as chlorophyll content, transpiration rate, stomatal conductance, leaf and tuber N, leaf area, leaf and tuber biomass, yield, and tuber number were observed for all varieties (Table 1). A significant variety × treatment interaction was observed in parameters such as leaf area, tuber N content, tuber biomass, and yield, the latter at the 5% significance level.

Highly significant differences (*p* ≤ 0.0001) were observed between treatments for manganese (Mn), carbon (C), potassium (K), C/N ratio, and ash content in leaves (Table 2). A highly significant variety x treatment interaction was observed for leaf manganese only. Variables related to tuber quality such as dry matter and reducing sugars showed no significant differences. Glutamic acid, aspartic acid, glycine, and arginine contents, as well as phenol content, in tubers showed highly significant differences (*p* ≤ 0.0001) among treatments (Table 3). The content of phenols in leaves, isoleucine, and antioxidant capacity showed differences at the 0.05% level (Table 3).

In relation to the physiological parameters evaluated in each of the three varieties, the chlorophyll content SPAD-502 chlorophyll meter (Konica Minolta, Osaka, Japan)in the N-stressed plants remained below that of non-stressed plants (Figure 1). The SPAD index is a measurement of leaf chlorophyll content and is obtained using a SPAD meter. It provides an indirect estimate of a plant’s N status, since chlorophyll levels are closely linked to N availability. Chlorophyll reduction reflects impaired nitrogen assimilation. Higher SPAD values typically indicate healthier, N-rich leaves, while lower values suggest N deficiency or stress.

The percentage of chlorophyll loss in these plants increased over time. The variety that showed the greatest loss of greenness was Monalisa with 27.7%, as opposed to 18.7% for Kennebec and 19.6% for Agria. Regarding the maximum photosynthetic efficiency of photosystem II (Fv/Fm), all control plants of the three varieties showed values of Fv/Fm higher than 0.81, with no significant differences between controls and plants subjected to N deficiency. However, CO_2_ assimilation (IRGAan), transpiration rate (IRGAe), and stomatal conductance (IRGAgs) were reduced in N-stressed plants of Kennebec and Agria varieties (Figure 2).

Nitrogen is essential for synthesizing Rubisco, the enzyme responsible for CO_2_ fixation during photosynthesis. CO_2_ assimilation declines, leading to stunted growth and lower biomass accumulation. Stomatal conductance reflects the opening of stomata, which is regulated by guard cells. Nitrogen supports ATP production and ion transport needed for guard-cell turgor. Under nitrogen stress conditions, stomatal conductance is expected to decrease. These changes cause lower transpiration rates, restricting nutrient transport.

Agria was the most stressed variety, with the greatest decrease in assimilation (33.46%) and transpiration (33.33%) and with the highest stomatal conductance decrease, reaching 35.29%, suggesting greater photosynthetic activity and gas exchange efficiency, although potentially associated with greater water losses. Kennebec, on the other hand, shows the lowest values in all the parameters evaluated (14.00% in assimilation losses, 4.54% in transpiration, and 4.87% in stomatal conductance), which could indicate a more conservative physiological strategy in response to water stress.

In the case of Monalisa, there is an increase in transpiration of 16.66% and in stomatal conductance of up to 6.60%, reflecting moderate activation of gas exchange. However, this response is accompanied by a loss in assimilation of 15.06%, which could indicate limited photosynthetic efficiency under the conditions evaluated. Taken together, these data suggest that Agria may be more suitable in environments with good water availability, while Kennebec and Monalisa may offer greater stability under stress conditions, albeit with different compromises in their physiological performance.

### 2.2. Biomass Changes

Biomass production, both fresh and dry weight, evaluated on day 49 of the cycle, at full flowering, was reduced by N deficiency in the three varieties evaluated (Figure 3). Nitrogen is a key component of chlorophyll, Rubisco, amino acids, nucleotides, and proteins needed for photosynthesis and cell proliferation. Thus, nitrogen deficit leads to production of lower carbohydrate content and reduced leaf expansion. However, it was a significant variable in Monalisa and Kennebec, but not in Agria. Leaf biomass was reduced by 40.25% and stem biomass by 37.18% in Agria, 23.63% and 18.72% in Kennebec, and 35.88% and 29.11% in Monalisa. Agria showed the highest total biomass of the three varieties but was also more affected by N reduction (38.85% reduction), while Monalisa showed a 33.40% reduction and Kennebec a 29.6% reduction. In all varieties, the percentage of dry matter accumulated in the tuber was higher in the stressed varieties than in the controls. In addition, leaf area was greater in the control plants, showing a greater decrease in Monalisa compared to Kennebec, with the least loss.

### 2.3. Nitrogen Content and C/N Ratio

The leaf C/N ratio was a highly significant parameter (0.001 level), with the controls showing lower values in all varieties due to N accumulation in the control plants. The ash percentage was also highly significant for treatments, being higher in N-deficient plants (Figure 4). In tubers, as in leaves, the C/N ratio showed lower values for the controls in the three varieties evaluated. Likewise, the percentage of total N in tubers was higher in the controls, but this was not the case for ash, which showed no differences between the control and stressed varieties (Figure 4).

Under stress conditions, a significant reduction in N content was observed in both leaves and tubers, accompanied by a marked increase in the C/N ratio. In leaf tissues, N levels declined by 27.54% in Agria, 35.38% in Monalisa, and 29.09% in Kennebec, while the C/N ratio rose by 33.15%, 50.49%, and 39.09%, respectively. This inverse relationship suggests a stress-induced imbalance in carbon and N metabolism. In tubers, N content decreased even more sharply—41.22% in Agria, 36.97% in Monalisa, and 39.50% in Kennebec—with corresponding C/N ratio increases of 71.89%, 63.06%, and 66.14%.

### 2.4. Yield Components

Yield components such as tuber weight per pot, number of tubers, and average tuber weight were measured at the end of the vegetative cycle. Agria was the variety that showed the lowest production and the lowest weight loss under stress conditions (15.73%), compared to Kennebec, which showed the highest percentage loss (29.22%), and Monalisa (26.58%), being a significant parameter in all three varieties. The number of tubers was higher in the control than in the stressed variety for Kennebec (a 22% decrease) and Agria (a 27,41% decrease), with no differences in the Monalisa variety (Figure 5). Mean tuber weight showed no significant difference between varieties.

### 2.5. Amino Acid and Phenolic Compound Profiles

We analyzed the phenols and essential and non-essential amino acids in both leaves and tubers for the three varieties tested. Nitrogen deficiency in potato alters amino acid and phenolic compound profiles by disrupting nitrogen assimilation, protein synthesis, and secondary metabolism, leading to stress-induced biochemical shifts. No significant differences were found in leaves for any of the varieties subjected to N deficiency stress compared to the control. In the tuber analysis, the contents of glutamine, asparagine, glycine, and arginine were higher in the controls than in the stressed plants in all varieties, with Agria showing the highest reduction in amino acid content under stress (a reduction of 31.32%), followed by Monalisa (a reduction of 20.63%) and Kennebec (a reduction of 8.36%). Tuber phenol content was a significant parameter, with Agria (38.98% of loss), as with amino acids, showing the highest phenol loss under N deficiency conditions, followed by Monalisa (34.74%) and Kennebec (30.04%) (Figure 6).

### 2.6. Correlation Analysis

Table 4 shows Pearson’s correlations between parameters, with some significance measured in N-deficient plants for the three varieties evaluated. Chlorophyll content, CO_2_ assimilation, transpiration, and stomatal conductance showed significant correlations. Likewise, manganese, microelements, asparagine, glutamic acid, glutamine, glycine, arginine, proline, and tuber phenol content were correlated with high significance. It was also observed that leaf microelements, gas exchange measurements (IRGAan, IRGAe, and IRGAgs), chlorophyll content, biomass, and reducing sugars correlated significantly with each other, and dry matter correlated significantly with tuber weight, as did vitamin C with leaf N content.

## 3. Discussion

This study has focused on identifying the different responses between control plants and plants subjected to N deficiency stress in three potato varieties, Agria, Kennebec, and Monalisa, based on the estimation of physiological and agronomic parameters.

In our study, we observed that nitrogen-deficient conditions led to a marked decline in stomatal conductance and chlorophyll fluorescence, consistent with findings from Guo et al. (2022), who reported significant reductions in dry matter and nitrogen accumulation under similar stress regimes [20].

Fertilized control plants showed higher chlorophyll levels, CO_2_ assimilation, transpiration, and stomatal conductance. In contrast, stressed plants exhibited a significant drop in chlorophyll across all three varieties. Lombardo et al. (2020) [7] observed that unfertilized plants showed lower SPAD values than those with normal fertilization.

Interestingly, despite the photosynthetic downturn, the efficiency of Photosystem II (PSII) remained essentially unaffected by the N deficit, suggesting that under moderate stress, the photochemical core remained stable. This partial decoupling of enzyme/biochemical constraints from purely photochemical ones has also been observed in other crops under N stress [21]. Among the three, Monalisa had the highest photosynthetic rate under both treatments, yet the lowest leaf and aerial biomass. However, it produced the highest tuber yield, suggesting that this genotype prioritizes reproductive development under stress conditions, rather than diverting resources to maintain vegetative growth.

One of the most consistent patterns across varieties was the shift in the C/N ratio under stress. The C/N ratio in both leaves and tubers showed lower values for the controls in all varieties, as control plants accumulate more N. An increase in the C/N ratio indicates that N and C acquisition are imbalanced, resulting in an apparent N deficiency, leading to sink limitation. Carbon and nitrogen metabolism are regulated by nutrient-sensitive pathways like TOR and SnRK1, which adjust gene expression and enzyme activity to balance growth and stress responses under changing C/N conditions. Recent studies have shown that SnRK1 and TOR also play central roles in modulating growth–defense trade-offs in plants, including potatoes. Under nitrogen stress, these signaling mechanisms prioritize carbon storage and stress resilience over vegetative expansion, which may explain why the stressed plants in our study accumulated more dry matter despite reduced nitrogen uptake [22,23].

Counterintuitively, stressed plants accumulated more dry matter than control plants. The C/N balance theory proposes that high N rates can limit plant growth, but carbon fixation (even at reduced rates) may continue, and plants tend to synthesize carbon-rich compounds such as starch [7]. In a study on *Citrus sinensis*, it was observed that N deficiency increased the C/N ratio in leaves, stems, and roots [24].

In leaves, non-essential amino acids represented over 80% of the total pool across all varieties and treatments. Cysteine B alone accounted for nearly 75%, highlighting its key role in sulfur and nitrogen assimilation. This dominance points to how sulfur and nitrogen assimilation systems remain tightly connected even under N limitation, perhaps reflecting the plant’s attempt to maintain core redox and sulfur metabolism under duress.

It should be noted that, in tubers, the Agria variety was the one that lost the most amino acids under N deficiency, being considered the most sensitive in these parameters when subjected to stress. Fertilized plants demonstrated elevated levels of asparagine, glutamine, and arginine—amino acids closely linked to N storage and transport. In contrast, stressed plants had more glutamic acid, crucial for carbon–nitrogen integration and transamination.

Under high-C/N-ratio conditions, plant metabolism prioritized the synthesis of metabolites (amino acids) with high C/N ratios, such as glutamic acid and proline, contrary to that observed in high-nitrate-fed plants [25]. This shift in metabolism represents an adaptive response in which cells prioritize carbon-rich amino acids to maintain cellular homeostasis and osmoprotection during nutrient scarcity. When the C/N ratio rises, cells tend to increase glutamate production. Glutamate not only acts as a building block for various other amino acids but also plays a key role in sustaining protein synthesis when N is in short supply [22].

At the enzymatic level, we did not directly assay enzyme activities, but our patterns echo trends seen in other studies. For instance, Chen et al. (2022), in a study with rice, observed that N reduction was related to the reduction in activity of enzymes related to N metabolism, such as nitrate reductase (NR), glutamine synthetase (GS), and glutamate synthase (GOGAT) [26]. These enzymes are central to the GS/GOGAT cycle, which assimilates ammonium into amino acids. All of them decreased in leaves and roots, which led to a decrease in total free amino acid content. Nitrogen deficit reduces ammonium availability for amino acid biosynthesis, disrupting transamination and altering amino acid and protein composition [27].

In *Brassica napus*, multi-omics studies show that nitrogen deficit disrupts amino acid synthesis and alters gene expression and metabolite profiles. It reduces N uptake genes and increases stress-related compounds like reactive oxygen species and organic acids [28]. Similar patterns have been documented in maize, where N deprivation led to elevated levels of soluble sugars and organic acids, alongside a decline in amino acid production [29].

One might expect that, under stress, starch and dry matter content would shift decisively—but we did not detect consistent differences between fertilized and N-deficient plants in those traits. Lombardo et al. (2020) reported that excessive nitrogen can paradoxically reduce dry matter and starch, likely because plants prioritize vegetative growth over storage compound synthesis [7]. In potato cultivars, Li et al. (2018) found that starch accumulation and the expression of starch synthase genes were highest under moderate N levels (150 kg ha^−1^) but declined at higher rates (300 kg ha^−1^), indicating that excessive N can suppress the enzymatic machinery responsible for starch biosynthesis [30]. In other words, beyond a threshold, extra nitrogen can hurt storage metabolism.

Guo et al. (2022) found that under nitrogen stress, potato plants reallocate resources toward tuber formation, with increased expression of starch and sucrose biosynthesis genes and elevated sucrose and glutamic acid levels [20]. In our study, Monalisa’s ability to sustain yield under stress may reflect a similar dynamic: prioritizing carbon fluxes toward tuber sink strength rather than vegetative maintenance.

On the nutritional side, vitamin C levels in tubers declined under N deficiency (a statistically significant drop in Agria). This is in accord with other studies that showed that N deficiency induces a reduction in vitamin C concentration in tubers [31,32]. In agreement, Fang et al. (2023) observed that the ascorbic acid content of tubers showed an increasing trend with increasing N application [33].

Phenolic content followed an intriguing bifurcation: in leaves, Monalisa under N deficiency showed an increase in total phenols (a canonical stress-induced antioxidant response) [34], but Agria and Kennebec did not show significant changes. In tubers, however, all three varieties lost phenolics under N stress, between 30% for the Kennebec variety and 39% for Agria, the latter being the one that showed the greatest decrease, as for other parameters evaluated. Previous works have shown a genotype effect on the nutritional composition of tubers and especially on the content of phenolic compounds [35,36,37].

Antioxidant activity was higher in fertilized plants, with significant differences observed only in Agria. An increase in fertilization results in an increment in the amounts of antioxidant compounds [7] that are determined by the skin, mainly concentrations of vitamin C, phenols, and carotenoids [38].

All three varieties showed yield losses under nitrogen deficit, with Agria experiencing the smallest reduction. Mean tuber weight was significant in Monalisa and the number of tubers in the Kennebec variety, both being higher in the fertilized control. This corresponds with other works that state that N deficiency has an impact on yield reduction in this crop [39,40].

The varietal cycle influenced nitrogen stress response, though not linearly. Agria, a long-cycle variety, had the smallest yield reduction (15.73%), despite marked declines in photosynthesis and amino acid levels—suggesting strong adaptive capacity through resource redistribution or stress tolerance. Its ability to accumulate higher biomass and sustain tuber quality points to a delayed but efficient stress compensation strategy.

In contrast, Kennebec, a medium–short-cycle variety, was the most affected, with a significant loss of production (29.22%), which could be attributed to its shorter time frame to compensate for N limitation during critical stages of development. Interestingly, Kennebec showed the most conservative physiological response, with minimal changes in gas exchange parameters. Kennebec’s shorter cycle may also restrict its capacity to recover from early nitrogen deficits, making it more vulnerable during critical growth stages.

Monalisa, a medium-sized-cycle variety, showed an intermediate response. It showed moderate reductions in photosynthetic efficiency and amino acid content, yet it maintained relatively high tuber yield. Interestingly, Monalisa increased stomatal conductance and transpiration under stress, suggesting an active gas exchange strategy that may support continued assimilating transport to tubers. This could explain its ability to sustain yield despite physiological limitations, indicating a genotype that prioritizes reproductive output over vegetative robustness.

In summary, nitrogen stress tolerance in potatoes depends not just on cycle length but on each genotype’s ability to balance resource allocation, stress signaling, and metabolic flexibility. A long-cycle variety like Agria has time to buffer stress, but only if its internal reallocation and signaling systems permit it. A short-cycle variety may not survive an early setback.

In correlational analyses under N deficiency, increased photosynthetic showed positive associations with reducing sugars in tubers, suggesting that residual photosynthesis under stress still contributes to C fluxes. Nitrogen use efficiency (NUE) is influenced by genetic traits, physiology, and environmental factors. In most crops, NUE is below 50% due to losses from volatilization, leaching, and microbial immobilization [41,42]. On the other hand, a high concentration of N usually has negative consequences on the quality of the tuber, such as a delay in physiological maturity and low dry matter, calcium, and magnesium content [43].

Recent studies reinforce and extend many of our observations, offering genetic, agronomic, and mechanistic insights that complement our results. For example, a 2025 GWAS study identified SNP markers linked to nitrogen stress traits like leaf area and tuber number, offering breeders tools to improve NUE through targeted selection [17]. Similarly, work by Zhang et al. (2024), using a combined transcriptomics, metabolomics, and physiological approach, has clarified how stolon/tuber formation is regulated under diverse nitrogen regimes, including deficiency [9]. Akkamis and Caliskan (2024) showed that combining nitrogen fertilization with optimal irrigation improves photosynthesis, leaf area, and chlorophyll content [44]. Their most effective treatment—intermediate nitrogen with full irrigation—supports our findings and highlights the need for genotype-specific nutrient strategies.

Recent field trials (Carruthers & Congreves 2025) in potato cultivars have shown that genotype (more than fertilizer rates per se) drives NUE and yield relationships—in effect, some varieties inherently make better use of soil N under diverse conditions [45]. Also, a field study comparing several ground-water nitrate environments showed that different potato cultivars respond differently to N fertilization in terms of yield, tuber size, and in-season N status, underlining the importance of genotype × environment interactions [46].

Collectively, these recent works support our conclusion that genotype, physiological traits, and efficient resource allocation underlie tolerance to N deficiency. Agria, though biochemically vulnerable under N stress, retains yield through robust internal reallocation and stress-compensatory pathways. Kennebec seems to avoid dramatic changes in leaf physiology, but this risk-averse stance may cost it in tuber yield when stress hits, and Monalisa actively shifts its physiology—opening stomata and maintaining gas exchange—potentially to sustain assimilate flow to tubers at the expense of leaf mass.

## 4. Materials and Methods

The trial was conducted in 2021. Sowing took place on 13 April, and harvesting on 21 July. 

### 4.1. Plant Material and Growth Conditions

Three potato varieties were selected from a previous screening for their differential ranges of productivity and maturity. Agria, Kennebec, and Monalisa are commercial varieties representative of those in Spanish markets, having different uses. The trial was carried out in the NEIKER greenhouse (Vitoria, Spain) during 2023. Plants grew at a temperature between 18 and 22 °C night/day, with 70% relative humidity, and supplemented with artificial light to obtain 16 h light/8 h dark. Twenty tubers per variety and treatment were planted in a completely randomized experimental design in 5 L pots with 50% peat-based growing medium (containing 140 mg/L N, 80 mg/L P_2_O_5_, and 190 mg/L K_2_O, Garlan Products Co. Ltd., Vitoria, Sapin) and 50% perlite. For each variety, ten pots were randomly selected to be subjected to N-deficit stress treatment (D), and the other 10 were irrigated normally (C).

The control pots were fertilized with 27% calcium ammonium nitrate twice during the growth cycle: once at emergence (11 days after planting), and at the onset of tuberization (29 days after planting, DAP). At each stage, 50% of the total fertilizer was applied, which was 1.14 g per pot. Although the recommended fertilizer application rate is over 500 kg/ha for maximum yield, we decided to apply 600 kg/ha, which translates to 2.28 g of fertilizer per 5 L.

Nitrogen-deficit stress pots were not fertilized. Each pot was watered frequently with 1 L of water and was placed in an individual tray so as not to lose substrate or mineral nutrients with drainage. Measurements were conducted between days 31 and 51 of the growth cycle, following the completion of both fertilization events in the control group. At the end of the vegetative cycle, tubers were harvested from all pots.

### 4.2. Physiological Parameters

The chlorophyll content, which is closely related to the greenness of the plant, was measured in each plant using a SPAD-502 chlorophyll meter (Konica Minolta, Osaka, Japan). Measurements were taken on 3 fully expanded leaves of each plant and in 4 plants of each variety and treatment on days 31, 36, and 43 of the cycle.

The maximum photosynthetic efficiency of photosystem II (ΦPSII) was determined in dark-adapted plants (Fv/Fm) using a fluorimeter (FluorPen FP 100, Photon Systems Instruments, Drasov, Czech Republic) on the last fully expanded leaves (LFELs) of three plants of each variety and treatment on days 37 and 46 after planting. These measures indicate the damage produced in photosystem II and the inhibition of photosynthesis.

IRGA (InfraRed Gas Analyzer, HORIBA, Kyoto, Japan)) was used to take measurements of the LFELs of three plants of each variety and treatment on days 37 and 46 after planting. CO2 assimilation (IRGAan), transpiration rate (IRGAe), and stomatal conductance (IRGAgs) were measured using an infrared gas analyzer (IRGA, ^®^Ciras-2, PP Systems International Inc., Amesbury, MA, USA) equipped with a universal photosynthesis cuvette [PLC(U)], as described by Saiz Fernández et al. (2017) [25]. This equipment is an open system, which means that photosynthesis and transpiration measurements are based on differences in CO_2_ and H_2_O in the air stream flowing through the leaf chamber.

### 4.3. Agronomic Parameters

The fresh and dry weights were measured on the last fully expanded leaf on day 38 after planting. Leaves were taken from 3 plants of each variety and treatment and immediately weighed to obtain fresh weight (FW); then, they were placed in an oven at 70 °C for 24 h to obtain dry weight (DW). Leaf area was estimated on the last fully expanded leaf on day 38 after planting. Three leaves per genotype and treatment were collected and scanned to obtain images and calculate the leaf area.

After the two fertilizations in the control plants, on day 49 after planting, 3 plants of each genotype and treatment were cut to measure the biomass. The aerial part of the plant was weighed to obtain fresh weight. Stems and leaves were separated from the aerial part and weighed. They were then placed in an oven at 120 °C until the plants were completely dry and weighed again to obtain dry weight.

Each pot was harvested 91 days after planting (DAP). Fresh tuber weight was measured for each pot (yield), and the total number of tubers was recorded. Dry matter content was measured in three tubers of each variety and treatment. Tubers were weighed immediately after harvest (FW). After 72 h at 80 °C, they were weighed again to obtain the dry weight (DW). The starch content was calculated with the following formula [47]:$$Starch=\left(\frac{DW}{FW}*100-6.0313\right)\times 10$$

Reducing sugar content in tubers was measured by spectrophotometry based on the reduction of dinitro salicylic acid. Three plants and two tubers per plant of each genotype and treatment were analyzed. The potatoes were peeled and mashed into homogeneous juice. A total of 0.3 g of the mixture was weighed, and 1 mL of distilled water and 2 mL of dinitro salicylic acid were added. Then the samples were heated at 100 °C in a water bath with stirring for 10 min. After this time, the samples were diluted with distilled water, and the absorbance was measured in a UV-VIS spectrophotometer GENESYS™ 50 (Thermo Fisher Scientific Inc., Waltham, MA, USA) at 546 nm. The percentage of reducing sugars was calculated as follows:$$\%reducing\;sugars=\left(absorbance-0.00385\right)\times 1.07893$$

### 4.4. Mineral, Carbon, Amino Acid, and Phenol Contents

The dry material was ground and sieved through a 0.12 mm stainless steel mesh. The ground material was homogenized and re-dried for at least 2 h at 80 °C before weighing out 0.5 g samples for analysis. Samples of the dried homogenate were wet-digested in a mixture of 1% HNO3 + 2% HClO_4_ (85:15, *v:v*) under a temperature gradient ranging from ambient to 190 °C for 12 h. The mineral contents (K, Ca, Mg, Na, P, S, Fe, Mn, Cu, and Zn) were determined and quantified by ICP-OES (5800 ICP-OES, Agilent Technologies, Santa Clara, CA, USA). Calibration standards were prepared from Certipur^®^ solutions (Merck, Darmstadt, Germany) for all minerals.

The N and carbon contents of subsamples (100 mg) of the dried homogenate were measured in an elemental analyzer (Leco CN828, Saint Joseph, MI, USA). Calibration was carried out using standard reference material Certificate of Traceability LECO^®^, Part No. B2141 Orchard Leaves.

The quantification of the amino acid content was carried out by means of an HPLC (Agilent Technologies 1260 Infinity II) system equipped with an autosampler (Agilent 1260 Vial Sampler G7129A) and diode array detector (Agilent 1260 FLD Spectra) in dry leaves and in a lyophilized tuber to determine the amounts of the main amino acids, as described by Minocha et al. (2004) [48]. Derivatization was carried out automatically using o-phthalaldehyde with 3-mercaptopropionic acid (OPA) for primary amino acids and 9-fluorenylmethylchloroformate (FMOC) for secondary amino acids. All reagents were sourced from Agilent Technologies Spain S.L., Madrid. For separation, an Advanced Bio AAA column (dimensions: 4.6 mm diameter; 2.7 μm particle size) from the same supplier was employed. The system operated at a temperature of 40 °C, with an injection volume of 1 μL. The mobile phases consisted of phase A: phosphate buffer at pH 8.2; and phase B: a mixture of methanol, acetonitrile, and water in a 45:45:10 ratio. Flow was set at 1.5 mL/min, and the following elution gradient was used: 0 min (2% B), 0.35 min (2% B), 13.4 min (57% B), 13.5 min (100% B), 15.7 min (100% B), 15.8 min (2% B), 18 (100% A). Detection was performed at 338 nm for OPA-derivatized amino acids and 262 nm for FMOC-derivatized ones. Amino acid concentrations were calculated using calibration curves specific to each compound. For each genotype and treatment condition, three biological samples were analyzed.

Total phenols were extracted by microwave-assisted extraction and quantified by the Folin–Ciocalteau spectrophotometric method in dry leaves and in lyophilized tubers. Three samples per genotype and treatment were analyzed. For this, it was necessary to carry out a wash of possible non-phenolic reducing interferents by means of solid-phase separation (SPE) as a step prior to the colorimetric reaction [49].

### 4.5. Antioxidant Capacity and Vitamin C

Tuber antioxidant activity was estimated as described by Tierno, R. (2017) [50], using the 2,2-diphenyl-1-picrylhydrazyl (DPPH) free radical method. It started with lyophilized material, using 0.1 mL of each sample, and 3.9 mL of DPPH• radical solution in MeOH: diH_2_O (70:30, *v/v*) (6 × 10^−5^ M) was added to each sample to initiate the reaction. The absorbance was measured at λ = 516 nm. The reaction time used for all DPPH analyses was 3 h. The blank used was MeOH: diH_2_O (70:30, *v/v*), and different trolox MeOH: diH_2_O (70:30, *v/v*) dilutions were used as standards (0.100, 200, 300, 400, and 500 μmol L^−1^). Antioxidant capacity was expressed in mol equivalents of trolox per kg of fresh weight (g ET kg^−1^ FW). Three samples per genotype and treatment were analyzed.

The quantitative determination of ascorbic acid in potato tubers was carried out using the High-Performance Thin-Layer Chromatography method, following Tierno et al.’s (2015) [36] method. A 2.5 mL solution of 4% oxalic acid was used as the standard. For the standard curve, a solution of 2.4 mL oxalic acid + 100 µL standard ASA solution 1 g/L was used. The units are micrograms/gr FW.

### 4.6. Statistical Analysis

Statistical analysis was performed using ANOVA in the environment R (v 3.3.2). All experimental data were averages of three replicates of each variety and treatment. Pearson’s correlation was used to determine the relationships between the parameters. Significant differences in the means within each stressed cultivar with respect to the control were compared at the 5% significance level (*p* ≤ 0.05), according to Tukey’s HSD test.

## 5. Conclusions

Based on the results obtained in this work, physiological parameters such as stomatal conductance and chlorophyll content can be measured rapidly and easily, making them highly useful for determining a response to stress derived from N deficiency and for identifying tolerant potato varieties. Agria showed greater resilience in maintaining tuber yield despite physiological declines, while Kennebec and Monalisa exhibited more pronounced reductions in nitrogen content and photosynthetic activity. These findings underscore the importance of genotype-specific evaluation for improving nitrogen use efficiency (NUE) in potato breeding programs. Beyond the limited number of varieties assessed, other constraints include the controlled greenhouse setting and the absence of long-term field validation. Indirect selection based on parameters associated with such stress can be a useful tool in potato breeding programs. Future research should expand these studies to other varieties, evaluating the most promising indirect parameters, and include molecular profiling to identify stable markers linked to NUE and stress tolerance, as well as validate the parameters studied in field trials.

## Figures and Tables

**Figure 1 plants-14-03237-f001:**
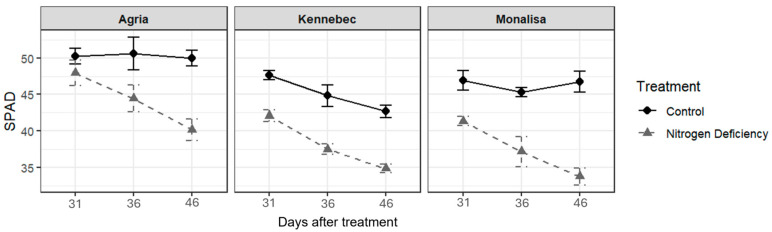
Leaf chlorophyll content (SPAD value, unitless index ranging from 0 to 60) measured in three potato varieties at three times (31: day 31 of the cycle; 36: day 36 of the cycle; 46: day 46 of the cycle) in control and N-deficiency-stressed plants.

**Figure 2 plants-14-03237-f002:**
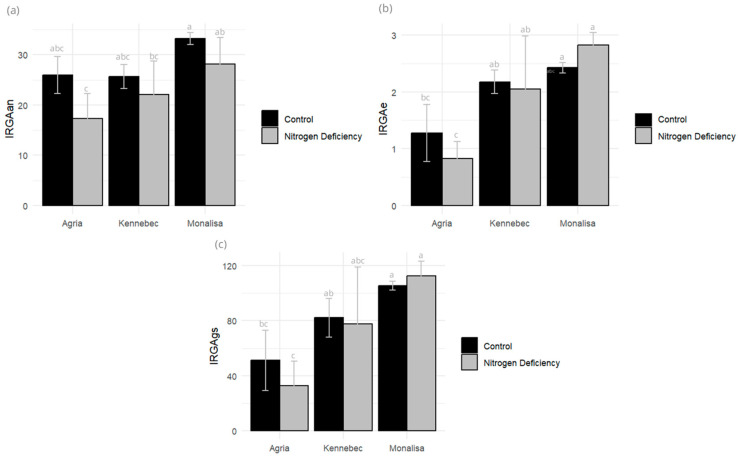
(**a**) CO_2_ assimilation rate—IRGAan (umol CO_2_·m^−2^·s^−1^); (**b**) transpiration rate—IRGAe (mmol H_2_O·m^−2^·s^−1^); and (**c**) stomatal conductance—IRGAgs (mmol m^−2^·s^−1^) measured in control and N-deficiency-stressed plants. Different letters indicate significant differences between the two treatments according to Tukey’s HSD test (*p* < 0.05).

**Figure 3 plants-14-03237-f003:**
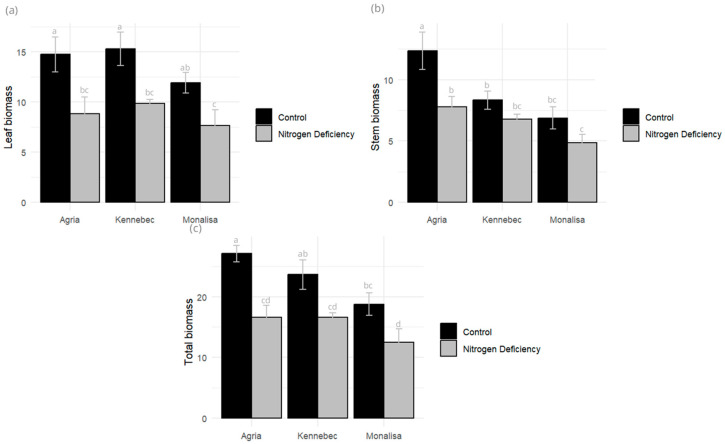
Effect of N deficiency on (**a**) leaf (g), (**b**) stem (g), and (**c**) total dry biomass (g) (leaf + stem biomass). Different letters indicate significant differences between the two treatments according to Tukey’s HSD test (*p* < 0.05).

**Figure 4 plants-14-03237-f004:**
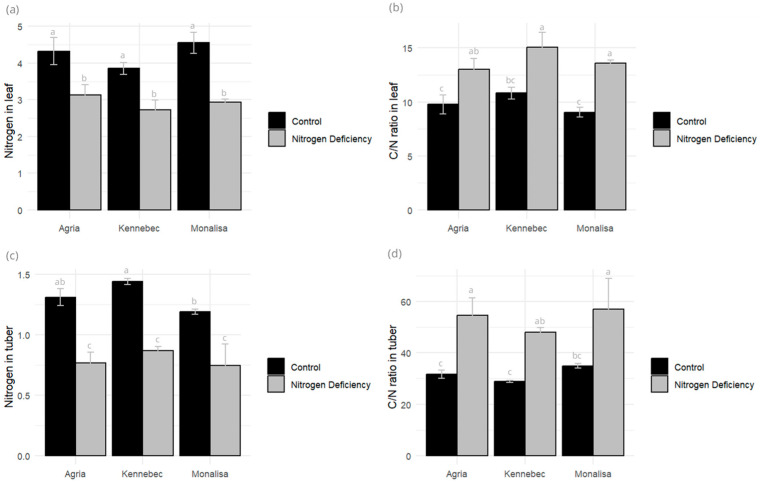
(**a**) Total N (%) in leaf, (**b**) C/N ratio LECO in leaf measured in control and N-deficiency-stressed plants, (**c**) total N (%) in tuber, and (**d**) C/N ratio LECO in tuber measured in control and N-deficiency-stressed plants. Different letters indicate significant differences between the two treatments according to Tukey’s HSD test (*p* < 0.05).

**Figure 5 plants-14-03237-f005:**
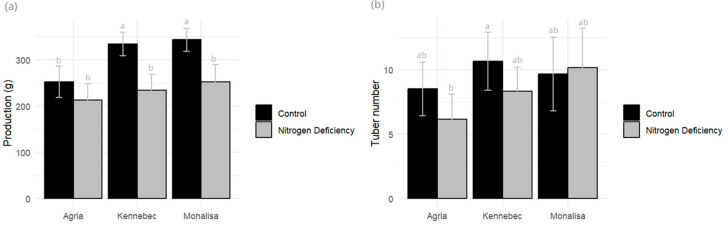
(**a**) Yield (g) and (**b**) tuber number (number of tubers) measured in control and N-deficiency-stressed plants. Different letters indicate significant differences between the two treatments according to Tukey’s HSD test (*p* < 0.05).

**Figure 6 plants-14-03237-f006:**
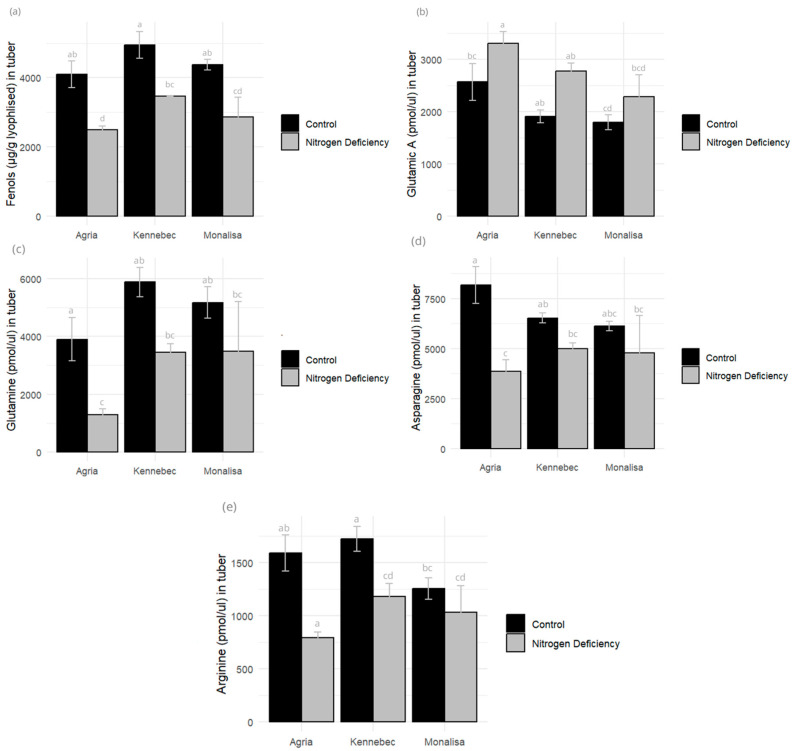
(**a**) Phenols (µg GAE/g FD), (**b**) glutamic acid (pmol/uL), (**c**) glutamine (pmol/uL), (**d**) asparagine (pmol/uL), and (**e**) arginine (pmol/uL) in tubers measured in control and N-deficiency-stressed plants. Different letters indicate significant differences between the two treatments according to Tukey’s HSD test (*p* < 0.05).

**Table 1 plants-14-03237-t001:** Analysis of variance of physiological and agronomic parameters in three potato varieties (V) under N deficiency and no N deficiency (Tr). Only significant parameters are shown. Chlorophyll content (SPAD value, unitless index ranging from 0 to 60), transpiration (IRGAe, mmol H_2_O·m^−2^·s^−1^), stomatal conductance (IRGAgs, mmol m^−2^·s^−1^), N in leaf (%), N in tuber (%), leaf area (cm^2^), yield (g), tuber number (number of tubers), leaf biomass (g), and tuber biomass (g).

	Chlorophyll Content	Transpiration	Stomatal Conductance	Nitrogen in Leaf	Nitrogen in Tuber	Leaf Area	Yield	Tuber Number	Leaf Biomass	Tuber Biomass
**Treatment (Tr)**	622.201 ***	177.670 ***	70.041	7.748 ***	44,909.38 **	153.125 ***	44,909.38 ***	36.380 **	122.200 ***	33.3472 ***
**Variety (V)**	86.640 ***	165.100 ***	8214.041 ***	0.398 *	14,971.33 **	130.172 ***	14,971.33 ***	25.330 **	12.600 *	26.8072 ***
**VxTr**	13.481	9.0804	192.791	0.103	4035.78 *	13.297 ***	4035.78 *	4.750	1.097	4.0372 *

*, **, *** significant at *p* = 0.05, *p* = 0.01, and *p* = 0.001, respectively.

**Table 2 plants-14-03237-t002:** Analysis of variance of LECO analysis and mineral content in three replicates of three potato varieties (V) under N deficiency and (Tr). The variables presented in the table were measured on the potato leaf. Mn l: manganese in leaf (mg/kg); Zn l: zinc in leaf (mg/kg); C l: carbon in leaf (mg/kg); Ca l: calcium in leaf (g/kg); K l: potassium in leaf (g/kg); Mg l: magnesium in leaf (g/kg); C/N ratio l: carbon-to-N ratio in leaf; Ash l: ash content in leaf (%).

	Mn l	Zn l	C l	Ca l	K l	Mg l	C/N Ratio l	Ash l
**Treatment (Tr)**	552.330 ***	57.501 *	6.125 ***	0.007	72.407 ***	4.603	72.320 ***	18.120 ***
**Variety (V)**	281.032 ***	75.660 *	1.131	21.590 **	129.841 ***	22.608 ***	5.024 *	3.343
**VxTr**	337.513 ***	0.490	0.171	0.325	10.964	1.184	0.700	18.120 ***

*, **, *** significant at *p* = 0.05, *p* = 0.01 and *p* = 0.001, respectively.

**Table 3 plants-14-03237-t003:** Analysis of variance of amino acids, phenols in leaf (Fen l; µg GAE/g FD), phenols in tuber (Fen t; µg GAE/g FD), and antioxidant capacity (CAox; % inhibition of DPPH radical at 516 nm) in three replicates of three potato varieties (V) under N deficiency and (Tr) measured by a spectrophotometer. The amino acids presented in the table are isoleucine in leaf (Ile l; pmol/uL), glutamic acid in tuber (Glu t; pmol/uL), aspartic acid in tuber (Asp t; pmol/uL), glycine in tuber (Gly t; pmol/uL), and arginine in tuber (Arg t; pmol/uL). Only significant variables are shown.

	Ile l	Glu t	Asp t	Gly t	Arg t	CAox	Fen l	Fen t
**Treatment (Tr)**	214.038 *	2,651,551.3 ***	27,157,123.4 ***	78,100.7 ***	1,314,052.1 ***	0.006 *	6,448,837.5 *	10,609,506,5 ***
**Variety (V)**	2.106	1,021,543.1 ***	668,194.1	186,990.1 ***	186,990.1 **	0.012 ***	22,494,377.3 ***	1,275,354.2 ***
**VxTr**	55.934	11,665.1	392,550.7 *	98,429.2 *	10.96	0.001	4,396,253.7 *	4704.2

*, **, *** significant at *p* = 0.05, *p* = 0.01, and *p* = 0.001, respectively.

**Table 4 plants-14-03237-t004:** Pearson’s correlation coefficients between significant parameters in N-deficient plants for the three varieties evaluated, using data from 30 samples. Chlorophyll content (SPAD value, unitless index ranging from 0 to 60), transpiration (IRGAe, mmol H_2_O·m^−2^·s^−1^), CO_2_ assimilation (IRGAan, umol CO_2_·m^−2^·s^−1^), stomatal conductance (IRGAgs, mmol m^−2^·s^−1^), leaf biomass (g), tuber biomass (g), leaf micronutrients (mg/kg), reducing sugars (%), dry matter (%), tuber number (number of tubers), and average tuber weight (g). *, ** significant at *p* = 0.05 and *p* = 0.01, respectively.

	Chlorophyll Content	Transpiration	CO_2_ Assimilation	Stomatal Conductance	Leaf Biomass	Tuber Biomass	Leaf Micronutrients	Reducing Sugars	Dry Matter	Tuber Number	Average Tuber Weight
**Chlorophyll content**	1	−0.394	−0.641 *	−0.599	−0.369	0.707 *	0.640 *	−0.719 *	−0.309	0.081	−0.308
**Transpiration**		1	0.873 **	0.921 **	0.602	−0.518	−0.647 *	0.621	−0.164	−0.029	0.001
**CO_2_ assimilation**			1	0.991 **	0.626	−0.734 *	−0.742 *	0.810 **	−0.028	0.174	−0.060
**Stomatal conductance**				1	0.647 *	−0.699 *	−0.742 *	0.775 *	−0.060	0.137	−0.044
**Leaf biomass**					1	−0.725 *	−0.845 **	0.698 *	−0.340	0.455	−0.377
**Tuber biomass**						1	0.826 **	−0.769 *	−0.077	−0.459	0.208
**Leaf micronutrients**							1	−0.698 *	−0.034	−0.298	0.107
**Reducing sugars**								1	−0.223	0.248	−0.237
**Dry matter**									1	−0.360	0.676 *
**Tuber number**										1	−0.849 **
**Average tuber weight**											1

## Data Availability

The original contributions presented in this study are included in the article. Further inquiries can be directed to the corresponding author(s).

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
