# Peer review of "Physiological Response to Nitrogen Deficit in Potato Under Greenhouse Conditions"

_plants, 2025, doi:10.3390/plants14213237_

Round 1
Reviewer 1 Report
Comments and Suggestions for Authors
The manuscript "Physiological response to nitrogen deficit in potato under greenhouse conditions" is logically structured and scientifically sound. The methodology is robust and supports the conclusions drawn. However, I have several minor suggestions to further improve the manuscript's clarity and impact.
Abstract
The sentence fragment on Line 19 ("while no additional.") must be completed.
Phrases such as "the one that lost more" and "showed the lowest values" are informal. Please use more precise, comparative language (e.g., "Agria exhibited the greatest reduction").
The result for the third variety is missing from the yield discussion; please include it for completeness.
The phrasing on Line 23 ("being Agria the one that lost more") is awkward and should be rewritten.
It would strengthen the abstract to include key quantitative data (e.g., percentage changes) for major findings.
Please add a sentence on future directions or implications at the end.
Introduction
Please cite recent literature on nitrogen deficiency and its role in plant physiology.
The research gap, hypothesis, and specific objectives should be more clearly and explicitly stated.
The abbreviation for nitrogen (N) is inconsistent; it is defined but the full word is used again later (e.g., Lines 56, 71, 82). Please use the abbreviation consistently throughout the manuscript.
Results
Please include the percentage change (increase or decrease) for key parameters to make the results more impactful and easier to interpret.
The quality of the figures needs improvement; please ensure they are high-resolution and clearly labeled.
Discussion
Improve the discussion by integrating more recent citations and presenting the findings in a more narrative, storytelling manner.
Clearly describe which parameters declined due to nitrogen deficit and explain the physiological mechanisms behind these responses.
Provide a comparative explanation for the differential responses observed among the three varieties, moving beyond simply citing literature to offering logical reasoning based on your results.
Ensure the discussion builds a clear and cohesive narrative that links your specific results to their broader scientific implications.
Methodology
Please provide details on the experiment's timing, including its start and end dates, as well as the total duration.
Author Response
Reviewer 1. The manuscript "Physiological response to nitrogen deficit in potato under greenhouse conditions" is logically structured and scientifically sound. The methodology is robust and supports the conclusions drawn. However, I have several minor suggestions to further improve the manuscript's clarity and impact.
Abstract
The sentence fragment on Line 19 ("while no additional.") must be completed.
Thank you for pointing out the incomplete sentence on Line 19. We have now completed the sentence to clarify .
Phrases such as "the one that lost more" and "showed the lowest values" are informal. Please use more precise, comparative language (e.g., "Agria exhibited the greatest reduction").
We have changed the phrase.
The result for the third variety is missing from the yield discussion; please include it for completeness.
The result of the third variety has now been introduced.
The phrasing on Line 23 ("being Agria the one that lost more") is awkward and should be rewritten.
The line 23 has been changed.
It would strengthen the abstract to include key quantitative data (e.g., percentage changes) for major findings.
We have included percentages for each variety in the abstract.
Please add a sentence on future directions or implications at the end.
We have added a sentence at he end of the abstract.
Introduction
Please cite recent literature on nitrogen deficiency and its role in plant physiology.
I have incorporated recent literature that highlights the physiological and molecular responses of plants to nitrogen deficiency.
The research gap, hypothesis, and specific objectives should be more clearly and explicitly stated.
I have revised the manuscript to more clearly and explicitly state and rewrite the research gap, hypothesis, and specific objectives.
The abbreviation for nitrogen (N) is inconsistent; it is defined but the full word is used again later (e.g., Lines 56, 71, 82). Please use the abbreviation consistently throughout the manuscript.
They have been modified in the article in all sections.
Results
Please include the percentage change (increase or decrease) for key parameters to make the results more impactful and easier to interpret.
The percentage changes for the SPAD value and IRGA parameters were already included in the article. Additionally, we’ve incorporated percentage changes for the other key parameters throughout the results section to enhance clarity and interpretability. Percentage changes for biomass, for nitrogen parameters for yield paramenters and for tuber phenols and aminoacids are now included in the article.
The quality of the figures needs improvement; please ensure they are high-resolution and clearly labeled.
I have ensured that all figures are set to 300 dpi for high-resolution quality, and they are correctly labeled in English.
Discussion
Improve the discussion by integrating more recent citations and presenting the findings in a more narrative, storytelling manner.
I have revised the discussion section to incorporate more recent and relevant literature, including studies published between 2022 and 2025.
Clearly describe which parameters declined due to nitrogen deficit and explain the physiological mechanisms behind these responses.
I have added a dedicated paragraph at the end of the discussion section that clearly summarizes the most significantly affected parameters under nitrogen deficit.
Provide a comparative explanation for the differential responses observed among the three varieties, moving beyond simply citing literature to offering logical reasoning based on your results.
I have included a comparative explanation in the discussion section that goes beyond literature citation by offering logical reasoning based on the results obtained in this study.
Ensure the discussion builds a clear and cohesive narrative that links your specific results to their broader scientific implications.
We have ensured that the revised discussion builds a clear and cohesive narrative by linking our specific findings to broader scientific implications.
Methodology
Please provide details on the experiment's timing, including its start and end dates, as well as the total duration.
We have added a sentence with the sowing and harvesting date of the trial

Reviewer 2 Report
Comments and Suggestions for Authors
Dear authors,
Please find my comments in the pdf file.

Author Response
Reviewer 2.
Abstract
Line 20 “highly significant differences” put the P value
I have integrated the p-value to the abstract.
Line 22 “SPAD values” put the SPAD values
I have included the SPAD values.
Line 30 “Solanum tuberosum L” not italic the L.
We have corrected the formatting.
Introduction
Line 33 “Solanum tuberosum L” not italic the L.
We have corrected the formatting.
Results
Table 1 Put the title above the table and what does the numbers represent? is this the mean value, F value, ss? please specify
I’ve already placed the title above the table 1
Line 102 “ Manganese, Carbon, Potassium,” Put abbreviations
The abbreviations for Manganese (Mn), Carbon (C), and Potassium (K) have already been included in line 102.”
Title of the table 2 K of Kg not in capital letter
The unit ‘kg’ in the title of Table 2 is already written with a lowercase ‘k’, as per standard notation.
Table 3 Put the title above the table
I’ve already placed the title above the table 3
Line 124 and line 125 explain more briefly SPAD index, this is too short
I introduced an expanded description in lines 128–131 to better contextualize its relevance to the observed physiological changes under stress conditions.
Figure 2, figure 3, figure 4, figure 5 and figure 6 put (a), (b) (c) and too small letters
I have corrected Figures 2 through 6 by properly labeling the subfigures with (a), (b), (c), etc., and ensuring the letters are now clearly visible and appropriately sized.
Table 4, put the N number of samples for correlation
N number of samples has been included in the table foot.
Discussion
Line 272-282 find more research for tuber plants
Additional recent research has been incorporated to strengthen the discussion on tuber plant responses to nitrogen deficiency.
Reviewer 3 Report
Comments and Suggestions for Authors
Dear Authors,
After a careful review of your manuscript, I would like to share some constructive comments and suggestions that may help strengthen your work:
Abstracts:
- The objective is stated, but the research gap is not well articulated. Why is this study necessary? Has the relationship between N deficiency and potato physiology/quality been poorly studied before?
Introduction:
- Several sections repeat ideas (e.g., excessive N causes vegetative growth at the expense of tubers is mentioned twice). This redundancy should be reduced.
- A sharper articulation of the gap is needed to justify the study.
- Improve logical flow: global → potato-specific → varietal differences → gap → aim.
- Clearly state the research gap and novelty.
- Ensure technical precision (units, mechanisms, terms like WUE/NUE).
- Strengthen objective statement (clear, concise, confident).
- Update and refine references.
- Improve language (reduce redundancy, avoid weak phrases).
Results:
- Reorganize into logical subsections for clarity.
- Report effect sizes with means ± SE instead of raw ANOVA outputs.
- Resolve contradictions in interpretation of variety responses.
- Avoid “data not shown” unless supplementary files are provided.
- Simplify correlation reporting, focusing on biologically meaningful results.
- Improve language for readability and scientific clarity.
Discussion:
- Simplify sentences, avoid repetition, and clarify contradictions.
- Provide mechanistic explanations for varietal differences.
- Reduce tangential literature; highlight direct potato-related studies.
- Report effect sizes, not just “higher/lower.”
- Emphasize novelty: link physiological/biochemical traits to breeding and stress tolerance.
- End with a strong synthesis, not background.
Materials and methods:
- Explicitly state the design used for this study.
- Provide a calculation table or reference for pot-to-field conversion and justify exceeding the recommended rate.
- Add light intensity values and clarify whether environmental conditions were monitored throughout the experiment.
- Physiological measurements: Standardize replication and report measurement timing.
- Agronomic parameters: Correct drying temperature and consider adding harvest index or tuber size distribution to strengthen agronomic analysis.
- Add details on calibration standards, derivatization steps, and detection conditions for HPLC.
- Include calibration standards and units (mg Trolox eq./g DW, mg ascorbic acid/100 g FW).
- Statistical analysis: Clarify biological vs. technical replicates. Consider adding multivariate analyses to strengthen interpretation.
Conclusion:
- Explicitly connect results to varietal performance and main findings.
- Emphasize broader implications for sustainable agriculture and NUE improvement.
- Expand on limitations beyond just the number of varieties.
- Provide concrete, specific future research directions.
- Strengthen the writing style to deliver a clear, impactful take-home message.
The language quality of the manuscript should be improved.
Author Response
Reviewer 3. Dear Authors,After a careful review of your manuscript, I would like to share some constructive comments and suggestions that may help strengthen your work:
Abstracts:
- The objective is stated, but the research gap is not well articulated. Why is this study necessary? Has the relationship between N deficiency and potato physiology/quality been poorly studied before?
The abstract has been revised to more clearly articulate the research gap.
Introduction:
- Several sections repeat ideas (e.g., excessive N causes vegetative growth at the expense of tubers is mentioned twice). This redundancy should be reduced.
One of the repeated statements has been removed to eliminate redundancy and improve clarity.
2. A sharper articulation of the gap is needed to justify the study.
I rewrite the research gap at the end of the introduction.
3. Improve logical flow: global → potato-specific → varietal differences → gap → aim.
The introduction has been restructured to improve logical flow.
4. Clearly state the research gap and novelty.
The introduction has been revised to clearly state both the research gap and the novelty of the study.
5. Ensure technical precision (units, mechanisms, terms like WUE/NUE).
I have revised the manuscript to ensure greater technical precision.
6. Strengthen objective statement (clear, concise, confident).
The objective statement has been revised to ensure it is clear, concise, and confidently presented.
7. Update and refine references.
I have incorporated new references.
8. Improve language (reduce redundancy, avoid weak phrases).
We have revised the Introduction section to enhance its logical structure and language.
Results:
1.Reorganize into logical subsections for clarity.
We have revised the manuscript structure to improve clarity and flow. The Results section has been reorganized into clearly defined subsections:
2.1 Physiological Responses to Nitrogen Deficiency
2.2 Biomass Changes
2.3 Nitrogen content and C/N ratio
2.4 Yield components
2.5 Amino Acid and Phenolic Compound Profiles
2.6 Correlation analysis
2. Report effect sizes with means ± SE instead of raw ANOVA outputs.
We consider that the three ANOVA tables provided offer sufficient information to interpret the results. In addition, the figures provided are more clarifying and allow for a clear reading of the results obtained by variety for each parameter.
3. Resolve contradictions in interpretation of variety responses
The percentages of loss and gain have been added to the results section to avoid contradictions, make the results more impactful and easier to interpret.
4. Avoid “data not shown” unless supplementary files are provided
All instances of “data not shown” have been removed .
5. Simplify correlation reporting, focusing on biologically meaningful results.
Non-significant associations were excluded from reporting to maintain clarity and relevance.
6. Improve language for readability and scientific clarity.
We have revised the Results section to enhance its logical structure and language.
Discussion:
- Simplify sentences, avoid repetition, and clarify contradictions.
I have simplified the sentence structures, removed repetitive content, and clarified any contradictions.
2. Provide mechanistic explanations for varietal differences.
I have included a comparative explanation for varietal differences in the discussion section.
- Reduce tangential literature; highlight direct potato-related studies.
I acknowledge that the literature directly focused on this type of potato-related study is limited. In response, I have made efforts to reduce tangential references and instead incorporated more recent and relevant studies that align closely with the scope of this research.
4. Report effect sizes, not just “higher/lower.”
I’ve addressed this point by incorporating specific percentage values to better illustrate the effect sizes.
5. Emphasize novelty: link physiological/biochemical traits to breeding and stress tolerance.
I have emphasized the novelty of the study by integrating a concluding paragraph at the end of the discussion section.
6. End with a strong synthesis, not background.
I have revised the conclusion to focus on a stronger synthesis of the key findings on our study.
Materials and methods:
- Explicitly state the design used for this study.
The design was a completely randomized design.
2. Provide a calculation table or reference for pot-to-field conversion and justify exceeding the recommended rate.
Based on a previous study carried out in the field during the years 2019 and 2020 and cited in the present work (Iribar et al. 2025), it was decided to use the same number of plants (20 per variety).
3. Add light intensity values and clarify whether environmental conditions were monitored throughout the experiment.
I have expanded the sentence where the conditions of the trial are specified.
4. Physiological measurements: Standardize replication and report measurement timing.
The number of plants (biological replicates) measured for each physiological parameter, as well as the day of the cycle on which they were measured, is already included in the text..
5. Agronomic parameters: Correct drying temperature and consider adding harvest index or tuber size distribution to strengthen agronomic analysis.
The drying temperatures are those already indicated in materials and methods. Regarding the size of the tubers, tubers were not separated by calibre.
6. Add details on calibration standards, derivatization steps, and detection conditions for HPLC.
A paragraph has been added about the standards, derivatization steps, and detection conditions used in HPLC.
7. Include calibration standards and units (mg Trolox eq./g DW, mg ascorbic acid/100 g FW).
A simple parapragh has been added to include calibration standards and units for Trolox and ascorbic acid.
8. Statistical analysis: Clarify biological vs. technical replicates. Consider adding multivariate analyses to strengthen interpretation.
In the description of each parameter measured, it is also specified whether the measurements were taken in the same plant (technical replication) or in different plants (biological replication).
A multivariate analysis has not been performed as we consider that this is not the objective of the study.
Conclusion:
1. Explicitly connect results to varietal performance and main findings.
I have explicitly connected the results to varietal performance
2. Emphasize broader implications for sustainable agriculture and NUE improvement.
I have addressed this point by emphasizing the broader implications of the findings for sustainable agriculture and nitrogen-use efficiency (NUE) improvement.
3. Expand on limitations beyond just the number of varieties.
Further limitations have been included.
4. Provide concrete, specific future research directions.
I have incorporated specific future research directions.
5. Strengthen the writing style to deliver a clear, impactful take-home message.
I have made an effort to strengthen the writing style.
Round 2
Reviewer 3 Report
Comments and Suggestions for Authors
Dear Authors,
Thank you for making significant improvements in the revised version. There are some minor points below that need to be improved:
- Lines 409 - 412, the information about temperature and relative humidity is repeated.
- Line 417: Please check the control and treatment. There are spelling mistakes.
- I recommend that you change the name of the treatment from nitrogen deficiency to nitrogen deficit or limited nitrogen.
- I guess the application of treatments is not well explained. Please rewrite this section.
Author Response
Dear Authors,
Thank you for making significant improvements in the revised version. There are some minor points below that need to be improved:
Lines 409 - 412, the information about temperature and relative humidity is repeated.
Thank you for your observation regarding the repetition of temperature and relative humidity. We have revised the section to eliminate redundancy.
Corrections have been marked in green in the manuscript.
Line 417: Please check the control and treatment. There are spelling mistakes.
We have revised the line, corrected the spelling and improved clarity.
I recommend that you change the name of the treatment from nitrogen deficiency to nitrogen deficit or limited nitrogen.
We have revised the manuscript and replaced “nitrogen deficiency” with “nitrogen deficit.” The term “nitrogen deficit” better captures the intended experimental condition.
I guess the application of treatments is not well explained. Please rewrite this section.
We have revised the section to clearly explain how the nitrogen deficit and control treatments were applied, including the timing and dosage behind the fertilization strategy.